# Adversarial Learning of General Transformations for Data Augmentation

Saypraseuth Mounsaveng[1,2], David Vazquez[2], Ismail Ben Ayed[1], and Marco Pedersoli[1]

[1]ÉTS Montréal, QC, Canada
[2]Element AI
*saypraseuth.mounsaveng.1@etsmtl.net, dvazquez@elementai.com, {ismail.benayed, marco.pedersoli}@etsmtl.ca*

## Abstract

Data augmentation (DA) is fundamental against overfitting in large convolutional neural networks, especially with a limited training dataset. In images, DA is usually based on heuristic transformations, like geometric or color transformations. Instead of using predefined transformations, our work learns data augmentation directly from the training data by learning to transform images with an encoder-decoder architecture combined with a spatial transformer network. The transformed images still belong to the same class but are new, more complex samples for the classifier. Our experiments show that our approach is better than previous generative data augmentation methods, and comparable to predefined transformation methods when training an image classifier.

## 1 Introduction

Convolutional neural networks have shown impressive results in visual recognition tasks. However, for proper training and good performance, they require large labeled datasets. If the amount of training data is small, data augmentation is an effective way to improve the final performance of the network (Hernández-García & König (2018); Perez & Wang (2017)). In images, data augmentation (DA) consists of applying predefined transformations such as flip, rotations or color changes (Krizhevsky et al. (2012); Ciresan et al. (2012)). This approach provides consistent improvements when training a classifier. However, the required transformations are dataset dependent. For instance, flipping an image horizontally makes sense for natural images, but produces ambiguities on datasets of numbers (*e.g.* 2 and 5).

Several recent studies investigate automatic DA learning as a method to avoid the manual selection of transformations. Ratner et al. (2017) define a large set of transformations and learn how to combine them. This approach works well however, as it is based on predefined transformations, it prevents the model from finding other transformations that could be useful for the classifier. Alternatively, Chongxuan et al. (2017) and Tran et al. (2017) generate new samples via a generative adversarial networks model (GAN) from the probability distribution of the data p(X), while Antoniou et al. (2018) learn the transformations of images, instead of generating images from scratch. These alternative methods show their limits when the number of training samples is low, given the difficulty of training a high-performing generative model with a reduced dataset. Hauberg et al. (2016) learn the natural transformations in a dataset by aligning pairs of samples from the same class. This approach produces good results on easy datasets like MNIST however, it does not appear to be applicable to more complex datasets.

Our work combines the advantages of generative models and transformation learning approaches in a single end-to-end network architecture. Our model is based on a conditional GAN architecture that learns to generate transformations of a given image that are useful for DA. In other words, instead of learning to generate samples from $p(X)$, it learns to generate samples from the conditional distribution $p(X|\hat{X})$, with $\hat{X}$ a reference image. As shown in Fig. 1(b), our approach combines a global transformation defined by an affine matrix with a more localized transformation defined by

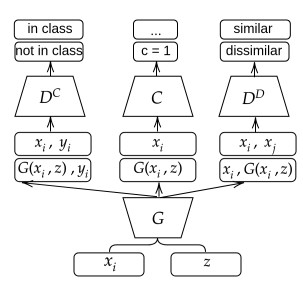

(a) Overview of model architecture.

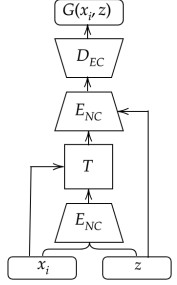

(b) Architecture of G.

Figure 1: **Our model** (a) A classifier $C$ receives augmented images from a generator $G$ constrained by two discriminators $D^C$ and $D^D$. The class discriminator $D^C$ ensures that the generated image $G(x_i, z)$ belongs to the same class $y_i$ as the input image $x_i$. The dissimilarity discriminator $D^D$ ensures that the transformed sample $G(x_i, z)$ is dissimilar from the input sample $x_i$ but similar to a sample $x_j$ from the same class. (b) Given an input image $x_i$ and a random noise vector $z$, our generator first performs a global transformation using a spatial transformer network followed by more localized transformations using a convolutional encoder-decoder network.

a convolutional encoder-decoder architecture. The global transformations are learned by an adaptation of spatial transformer network (STN) (Jaderberg et al. (2015)) so that the entire architecture is differentiable and can be learned with standard back-propagation. In its normal use, the purpose of STN is to learn how to transform the input data, so that the model becomes invariant to certain transformations. In contrast, our approach uses STN to generate augmented samples in an adversarial way. With the proposed model we show that, for optimal performance, it is important to jointly train the generator of the augmented samples with the classifier in an end-to-end fashion. By doing that, we can also add an adversarial loss between the generator and classifier such that the generated samples are difficult, or adversarial, for the classifier.

To summarize, the contributions of this paper are: i) We propose a DA network that can automatically learn to generate augmented samples without expensive searches for the optimal data transformations; ii) Our model trains jointly with a classifier, is fully differentiable, trainable end-to-end, and can significantly improve the performance of any image classifier; iii) In low-data regime it outperforms models trained with strong predefined DA; iv) Finally, we notice that, for optimal performance, it is fundamental to train the model jointly with the image classifier.

## 2 OUR APPROACH

We propose a GAN based architecture that learns to augment training data for image classification. As shown in Fig. 1(a), this architecture involves four modules: a generator to transform an input image, two discriminators and a classifier to perform the final classification task. In Fig. 1(b) we show the structure of the generator. Instead of generating a new image, as in most GAN models, our generator learns to transform the input image. Our intuition is that learning an image transformation instead of learning a mapping from noise to image is an easier task in low data regime.

Given an input image $x_i$ and a noise vector $z$, $E_{NC}$ converts them into a small representation that is passed to $T$ to generate an image and noise dependent affine transformation (similar to spatial transformer networks STN) of the original image. This transformed image is then passed to a U-Net network (Ronneberger et al. (2015)) represented by $E_{NC}$ and $D_{EC}$. While in the original paper STN was used for removing invariances from the input data, the proposed model generates samples with transformations that can help to learn a better classifier.

The generator is supported by two discriminators. The first one, the class discriminator $D^C$, ensures that the generated sample belongs to the same class as the input sample. The second one, the dissimilarity discriminator $D^D$, forces the transformed sample to be dissimilar to the input sample but similar to a sample from the same class. This is necessary to prevent the generator from learning the identity transformation. More details about the different networks and the loss functions used to train the model can be found in Appendices A and B.

## 3 EXPERIMENTS

In this section, we present several experiments to better understand our model and compare it with the state-of-the-art in automatic DA. We test our approach on MNIST, Fashion-MNIST, SVHN, and CIFAR-10, both in full dataset and low-data regime.

### 3.1 COMPARISON WITH STANDARD DATA AUGMENTATION

In this series of experiment, we compare the efficiency of the DA learned by our model to a heuristically chosen DA. We consider two different levels of DA. *Light DA* refers to random padding of 4 pixels on each side of the image, followed by a crop back to the original image dimensions. *Strong DA* adds to the previous transformations also rotation in range [-10, 10] degrees and scaling, with factor in range [0.5, 2]. For CIFAR10, DA also includes a horizontal image flip.

In a first experiment, we compare the accuracy of the baseline classifier, the baseline with DA, and our DA model while increasing the number of training samples. In our model, the classifier is trained jointly with the generator.

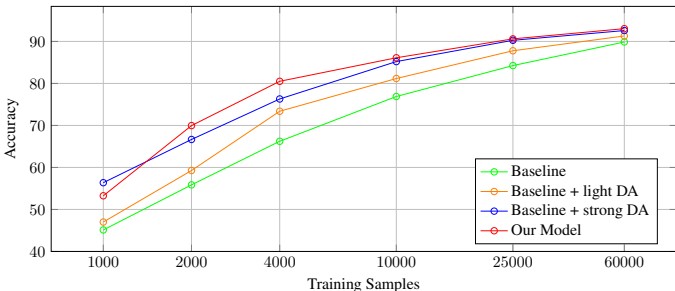

Figure 2: **Classification Accuracy vs number of training samples on CIFAR10.** Our method is effective when the number of samples is reduced. However for too few samples, normal DA is still slightly better.

In Fig. 2, we notice that for very few samples (1000) the predefined DA is still better than our approach. This is probably because when the training dataset is too small, the generator produces poor samples that are not helpful for the classifier. When the number of samples increases, our approach obtains a much better accuracy than strong DA. For instance, at 4000 training samples, the baseline obtains an accuracy of 66%, the predefined DA approach 76%, and our model 80.5%, thus a net gain of 14 points compared to the baseline and 4 points compared to strong DA model. If we add more examples, the gap between our learned DA and the strong DA tends to reduce. With the full dataset we reach about a half point better than the strong DA.

In a second experiment, we compare different types of DA on four datasets with a reduced number of samples.

| Method | MNIST 550 | FMNIST 550 | SVHN 1000 | CIFAR10 4000 |
|---|---|---|---|---|
| Baseline | 90.81 | 79.02 | 79.55 | 66.73 |
| Baseline + light DA | 97.55 | 78.96 | 84.48 | 74.76 |
| Baseline + strong DA | 98.50 | 80.37 | 84.33 | 77.74 |
| Our best model | 98.61 | 82.43 | 86.07 | 80.5 |

Table 1: **Comparison with DA on different datasets**. In low data regime, our model performs better than light and strong DA on the four considered datasets.

As shown in Tab. 1, our best model is always performing better than *light DA* and *strong DA*. This means that our DA model learns transformations that are more useful for the final classifier. Notice that in FMNIST *light DA* decreases performance of the final classifier. This suggests that DA is dataset dependent and transformations producing useful new samples in some domains might not be usable in others.

## 3.2 JOINT TRAINING

In this experiment, we report the performance of our method on *Joint training* and *Separate training* and compare them with a *Baseline* model trained without DA. In *Joint training* the generator of augmented images and the classifier are trained simultaneously in an end-to-end training. In *Separate training* instead, the generator is first trained to generate augmented images, and these images are then used as DA to improve the classifier.

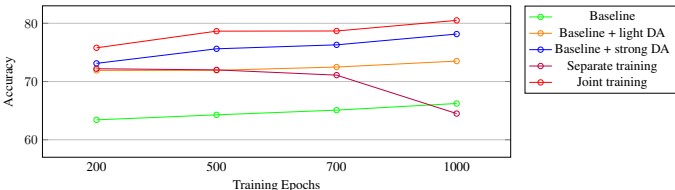

Figure 3: **Classification Accuracy** over epochs on 4000 samples of CIFAR10. We compare a baseline classifier with no DA and our joint training with a separate DA training in which samples are collected after 200, 500, 700 and 1000 training epochs.

In Fig. 3, we notice the different behavior of the two methods. In the early phase of training, at epoch 200, both *Separate training* (beige bar) and the *Joint training* (red bar) perform above 70%, whereas *Baseline* has a much lower accuracy. However, with additional training epochs, the performance of *Separate training* decreases while *Baseline* and *Joint training* accuracies increase. We believe that for good performance in DA it is not just about generating plausible augmented samples, but also about generating the right samples at the right moment, as in curriculum learning (Bengio et al. (2009)). From this experiment, it seems clear that for DA based on generic transformations (in contrast to predefined transformations as in (Ratner et al. (2017)), the joint training of the generator and the classifier is important for optimal performance. This can explain why DAGAN (Antoniou et al. (2018)) seems to work only with a very reduced set of examples.

## 3.3 COMPARISON WITH STATE OF THE ART

| Method | Model | MNIST 550 | CIFAR10 4000 | CIFAR10 Full |
|---|---|---|---|---|
| Baseline | ConvNet | 90.81 | 66.23 | 89.88 |
| Bayesian DA Tran et al. (2017) | ResNet18 | - | - | 91.0 |
| DADA Zhang et al. (2019) | ResNet56 | - | 79.3 | - |
| TANDA Ratner et al. (2017)(MF) | ResNet56 | 96.5 | 79.5 | 94.4 |
| TANDA Ratner et al. (2017)(LSTM) | ResNet56 | 96.7 | 81.5 | 94.0 |
| Our model | ConvNet | 96.0 | 80.5 | 93.0 |

Table 2: **Automatic DA Methods.** We compare the accuracy of our model with other methods performing automatic DA on MNIST and CIFAR10.

In Tab. 2, we compare our method with other approaches for automatic DA. Compared with *TANDA* (Ratner et al. (2017)), our method obtains slightly lower accuracies. However, *TANDA* is based on the selection of multiple predefined transformations. This means that its learning is reduced to a set of manually selected transformation, which, we believe, reduces the search space and facilitates the task. Also, *TANDA* uses an additional standard DA based on image crop, while our method does not need any additional DA.

On the other hand, our method compares favorably to *Bayesian DA* (Tran et al. (2017)) and *DADA* (Zhang et al. (2019)), both based on GAN models with a larger neural network for the classifier. This shows that our combination of global and local transformations helps to improve the final performance of the method.

## 4 CONCLUSION

In this work, we have presented a new approach for improving the learning of a classifier through an automatic generation of augmented samples. The method is fully differentiable and can be learned end-to-end. In our experiments, we have shown several elements contributing to an improved classification performance. First, the generator and the classifier should be trained jointly. Second, the

combined use of global transformations with STN and local transformation with U-Net is essential to reach the highest accuracy levels. For future work, we want to include more differentiable transformations such as deformations and color transformations and evaluate how these additional sample augmentations affect the final accuracy.

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

## APPENDIX A   LOSS FUNCTIONS

**Generator.**   The role of the generator G, is to learn which transformations to input images are the most useful to train the classifier. In our intuition, learning an image transformation instead of learning a mapping from noise to image to generate additional samples, is an easier task in low data regime. Given an input image $x_i$ and a noise vector $z$, the generator $G$, composed of an encoder $E_{NC}$, a decoder $D_{EC}$ and an affine transformer $T$, learns a transformation of the image that helps to train the classifier $C$. So, the transformation can be formulated as:

$$G(x_i, z) = D_{EC}(E_{NC}(T(x_i, E_{NC}(x_i, z)), z)) \tag{1}$$

The loss of the generator can be formulated as:

$$\begin{aligned}
\mathcal{L}_G = &-\alpha \mathbb{E}_{x_i, y_i \sim p_{data}, z \sim p_z} \left[ \log \left( D^C(G(x_i, z), y_i) \right) \right] \\
&-\beta \mathbb{E}_{x_i \sim p_{data}, z \sim p_z} \left[ \log \left( D^D(x_i, G(x_i, z)) \right) \right] \\
&-\gamma \mathbb{E}_{x_i, y_i \sim p_{data}} \left[ \log \left( 1 - C_{y_i}(G(x_i)) \right) \right]
\end{aligned} \tag{2}$$

where $x_i$ is an input image with class label $y_i$, $G(x_i, z)$ is a transformation of the sample $x_i$ and a noise vector $z$. $D^C$, $D^D$ and $C$ are respectively the class discriminator, the dissimilarity discriminator and the classifier and will be addressed in the following paragraphs. $\alpha$, $\beta$ and $\gamma$ are hyper-parameters introduced to balance the three loss terms and stabilize the training of the model.

In the first term of the loss function, the probability that a pair (transformed sample, true label) is classified as fake is minimized. In the second term, the dissimilarity between the original image and the transformed image is maximized. Finally, in the third one, the predicted probability of the real label for the transformed sample is minimized in order to make the classifier robust against adversarial samples.

**Class discriminator.**   During training, the generator is supported by two discriminators. The first one is referred to as class discriminator $D^C$. It ensures that the generated image belongs to the same class as the original image. $D^C$ takes as input an image and a class label and outputs the probability of the image to belong to that class. Its loss function is:

$$\begin{aligned}
\mathcal{L}_{D^C} = &-\mathbb{E}_{x_i, y_i \sim p_{data}} \left[ \log \left( D^C(x_i, y_i) \right) \right] \\
&-\mathbb{E}_{x_i, y_i \sim p_{data}, z \sim p_z} \left[ \log \left( 1 - D^C(G(x_i, z), y_i) \right) \right]
\end{aligned} \tag{3}$$

The first term increases the probability of $D^C$ for a real sample $x_i$ of class $y_i$, whereas the second term reduces $D^C$ for a generated sample $G(x_i, z)$ of the same class. In this way the discriminator learns to distinguish between real and generated samples of a certain class.

**Dissimilarity discriminator.**   The second discriminator, called dissimilarity discriminator $D^D$, ensures that the generated sample is as different as possible from the original sample. $D^D$ takes a pair of samples as input, and outputs a dissimilarity score between the two samples, ranging between 0 and 1, where 0 means that the two samples are identical. Its loss function can be formulated as:

$$\begin{aligned}
\mathcal{L}_{D^D} = &-\mathbb{E}_{x_i, x_j \sim p_{data}} \left[ \log \left( D^D(x_i, x_j) \right) \right] \\
&-\mathbb{E}_{x_i \sim p_{data}, z \sim p_z} \left[ \log \left( 1 - D^D(x_i, G(x_i, z)) \right) \right]
\end{aligned} \tag{4}$$

In the first term of the loss function, the dissimilarity between the original sample and another true sample from the same class is maximized, whereas in the second one, the dissimilarity between a true sample and the corresponding transformed sample is minimized.

**Classifier.**   The image classifier $C$ is trained jointly with the generator and the two discriminators. $C$ is fed with real samples as well as augmented samples, *i.e.* samples transformed by $G$. Its loss function is:

$$\begin{aligned}
\mathcal{L}_C = &-\mathbb{E}_{x_i, y_i \sim p_{data}} \left[ \log \left( C_{y_i}(x_i) \right) \right] \\
&-\mathbb{E}_{x_i, y_i \sim p_{data}, z \sim p_z} \left[ \log \left( C_{y_i}(G(x_i, z)) \right) \right]
\end{aligned} \tag{5}$$

In the first term of the loss function, the cross entropy loss between the predicted labels of the true samples and the true label distribution is minimized. In contrast, the cross entropy loss between the predicted labels of the transformed samples and the true label distribution is minimized in the second term.

**Global Loss.** Finally, we want to minimize a global loss to find the optimal parameters for the generator, discriminators and classifier. This loss is defined as:

$$\mathcal{L} = \mathcal{L}_G + \mathcal{L}_{D^C} + \mathcal{L}_{D^D} + \mathcal{L}_C \tag{6}$$

During optimization, we sequentially minimize a mini-batch of each loss. Notice that $\mathcal{L}_G$ tries to minimize the cross-entropy of $D^C$ of the transformed samples $G(x_i, z)$, while $\mathcal{L}_{D^C}$ tries to minimize $1 - D^C$. The same also for $D^D$ and $C$. This is not a problem, in fact this shows that the defined loss is adversarial, in the sense that generator and discriminator//classifier fight to push the losses in different directions. If the optimization is tuned properly, this mechanism generates augmented samples that are good for training the classifier, *i.e.* samples that belongs to the right class but are close to the decision boundaries.

## APPENDIX B    IMPLEMENTATION DETAILS

In all our experiments, we apply a basic pre-processing to the images, which consists in subtracting the mean pixel value, and then dividing by the pixel standard deviation. The generator is a combination of a STN (Jaderberg et al. (2015)) module followed by a U-Net (Ronneberger et al. (2015)) network. The generator network takes as input an image and a Gaussian noise vector (100 dimensions), which are concatenated in the first layer of the network. The three parameters $\alpha, \beta$ and $\gamma$ of the generator loss are estimated on a validation set. For the class discriminator $D^C$, we use the same architecture as in Dai et al. (2017). The network is adapted to take as input an image and a label (as a one hot vector). These are concatenated and given as input to the first layer of the architecture. For the dissimilarity discriminator $D^D$, we also use the same architecture. The network is adapted to take as input a pair of images, which are concatenated in the first layer of the architecture. For the classifier, we use the architecture used in Dai et al. (2017). We use Adam as optimization method.

**Training parameters**    To train our model, we use following values for the optimization parameters. Generator: Adam optimizer with a initial learning rate of 0.0005, a $\beta_1$ value of 0.5 and a $\beta_2$ value of 0.999. Class Discriminator: Adam optimizer with a initial learning rate of 0.0005, a $\beta_1$ value of 0.5 and a $\beta_2$ value of 0.999. As balance factor (see Sec. 2), we use as value for $\alpha$ 0.1 for MNIST, 1 for SVHN and 0.1 for CIFAR10. Similarity Discriminator: Adam optimizer with a initial learning rate of 0.0005, a $\beta_1$ value of 0.5 and a $\beta_2$ value of 0.999. As balance factor (see Sec. 2), we use as value for $\beta$ 0.05 for MNIST, 1 for SVHN and 0.05 for CIFAR10. Classifier: Adam optimizer with a initial learning rate of 0.006, a $\beta_1$ value of 0.5 and a $\beta_2$ value of 0.999. As balance factor (see Sec. 2), we use as value for $\gamma$ 0.005 for MNIST, 0.0005 for SVHN and 0.001 for CIFAR10.

**Detailed architectures**    In Tab. 3 we show the details of the classifier $C$, in Tab. 4 and Tab. 5, the details of respectively the two discriminators $D^C$ and $D^D$. In Tab. 6, we can see the details for the generator $G$.

**Classifier C**

| Input 32x32 Image |
| --- |
| 3x3 conv. 96 LReLU(0.2) |
| 3x3 conv. 96 LReLU(0.2) |
| 3x3 conv. 96 LReLU(0.2), 0.5 dropout |
| 3x3 conv. 192 LReLU(0.2) |
| 3x3 conv. 192 LReLU(0.2) |
| 3x3 conv. 192 LReLU(0.2), 0.5 dropout |
| 3x3 conv. 192 LReLU(0.2) |
| 3x3 conv. 192 LReLU(0.2) |
| 3x3 conv. 192 LReLU(0.2) |
| MLP 10 unit, sigmoid |
| 10-class Softmax |

Table 3: **Details of C network**.

**Discriminator $D^C$**

| Input 32x32 Image | Input One-hot class representation |
| --- | --- |
| 3x3 conv. 48 LReLU(0.2) | 32x32 deconv. 48 LReLU(0.2) |
| 3x3 conv. 96 LReLU(0.2) | |
| 3x3 conv. 96 LReLU(0.2), 0.5 dropout | |
| 3x3 conv. 192 LReLU(0.2) | |
| 3x3 conv. 192 LReLU(0.2) | |
| 3x3 conv. 192 LReLU(0.2), 0.5 dropout | |
| 3x3 conv. 192 LReLU(0.2) | |
| 1x1 conv. 192 LReLU(0.2) | |
| 1x1 conv. 192 LReLU(0.2), 0.5 dropout | |
| MLP 1 unit, sigmoid | |

Table 4: **Details of $D^C$ network**.

**Discriminator $D^D$**

| Input 32x32 Image | Input 32x32 Image |
| --- | --- |
| 3x3 conv. 48 LReLU(0.2) | 3x3 conv. 48 LReLU(0.2) |
| 3x3 conv. 96 LReLU(0.2) | |
| 3x3 conv. 96 LReLU(0.2), 0.5 dropout | |
| 3x3 conv. 192 LReLU(0.2) | |
| 3x3 conv. 192 LReLU(0.2) | |
| 3x3 conv. 192 LReLU(0.2), 0.5 dropout | |
| 3x3 conv. 192 LReLU(0.2) | |
| 1x1 conv. 192 LReLU(0.2) | |
| 1x1 conv. 192 LReLU(0.2), 0.5 dropout | |
| MLP 1 unit, sigmoid | |

Table 5: **Details of $D^D$ network**.

**Generator G**

| Input 32x32 Image | 100-dim noise vector |
|---|---|
| 3x3 conv. 32,batchNorm,LReLU(0.2) | - |
| 3x3 conv. 32,batchNorm,LReLU(0.2) | 32x32 deconv. 32, LReLU(0.2) |

| |
|---|
| Down STN |
| 2x2 max-pooling |
| 3x3 conv. 64,batchNorm,LReLU(0.2) |
| 3x3 conv. 128,batchNorm,LReLU(0.2) |
| 2x2 max-pooling |
| 3x3 conv. 256,batchNorm,LReLU(0.2) |
| 3x3 conv. 256,batchNorm,LReLU(0.2) |
| 2x2 max-pooling |
| 3x3 conv. 512,batchNorm,LReLU(0.2) |
| 3x3 conv. 512,batchNorm,LReLU(0.2) |
| 2x2 max-pooling |
| 3x3 conv. 1024,batchNorm,LReLU(0.2) |
| 3x3 conv. 1024,batchNorm,LReLU(0.2) |
| MLP 32 unit, ReLU |
| MLP 6 unit |
| Down U-Net |
| 2x2 max-pooling |
| 3x3 conv. 64,batchNorm,LReLU(0.2) |
| 3x3 conv. 128,batchNorm,LReLU(0.2) |
| 2x2 max-pooling |
| 3x3 conv. 256,batchNorm,LReLU(0.2) |
| 3x3 conv. 256,batchNorm,LReLU(0.2) |
| 2x2 max-pooling |
| 3x3 conv. 512,batchNorm,LReLU(0.2) |
| 3x3 conv. 512,batchNorm,LReLU(0.2) |
| 2x2 max-pooling |
| 3x3 conv. 1024,batchNorm,LReLU(0.2) |
| 3x3 conv. 1024,batchNorm,LReLU(0.2) |
| Up U-Net |
| 3x3 conv. 512,batchNorm,LReLU(0.2) |
| 3x3 conv. 512,batchNorm,LReLU(0.2) |
| 3x3 conv. 256,batchNorm,LReLU(0.2) |
| 3x3 conv. 256,batchNorm,LReLU(0.2) |
| 3x3 conv. 128,batchNorm,LReLU(0.2) |
| 3x3 conv. 128,batchNorm,LReLU(0.2) |
| 3x3 conv. 64,batchNorm,LReLU(0.2) |
| 3x3 conv. 64,batchNorm,LReLU(0.2) |
| 1x1 conv. (3 for color, 1 for grayscale),batchNorm,LReLU(0.2) |

Table 6: **Details of G network**.

## APPENDIX C    EXAMPLES OF TRANSFORMATIONS

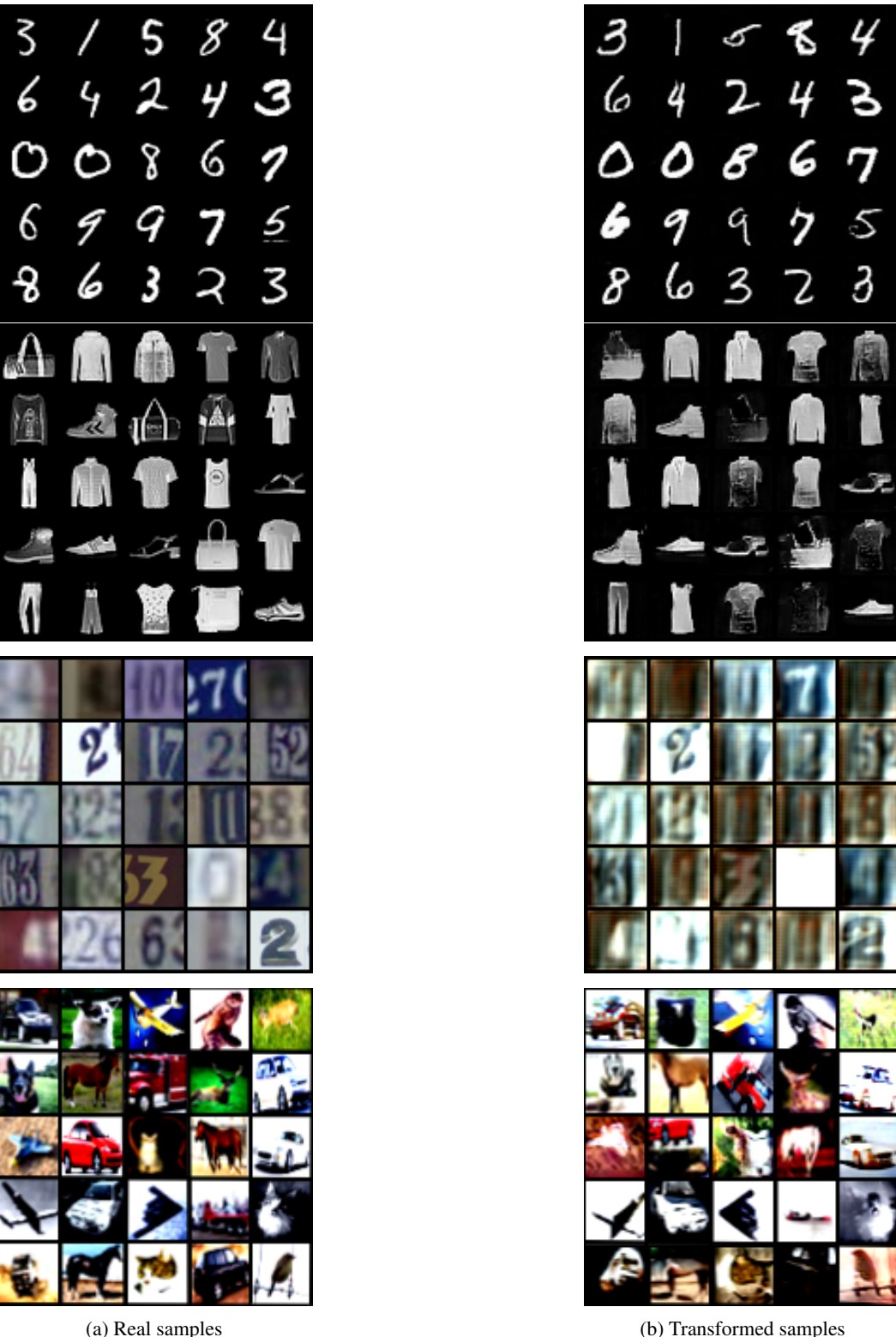

(a) Real samples  (b) Transformed samples

Figure 4: **Real and transformed images** from MNIST, Fashion-MNIST, SVHN and CIFAR10. Our approach learns to apply the right transformations for each dataset. For instance on MNIST and Fashion-MNIST there is no flip, nor zoom, because not useful, while on SVHN zoom is often used and on CIFAR10, both zoom, flip and color changes are applied.

