# OpenReview forum: "Adversarial Learning of General Transformations for Data Augmentation"
_ICLR.cc/2019/Workshop/LLD — LLD 2019_

### Official Review · AnonReviewer2 · 2019-04-03
**Easy to follow, Interesting approach, Results not so decisive**

**Rating:** 3
**Confidence:** 3

**Review:**

The authors of this work propose and evaluate a scheme for data augmentation (DA) that is based on rigid and non-rigid transformations of the training samples. The proposed approach is based on a generative adversarial training paradigm that is constrained by two discriminators that regularize the generated(-transformed) images to conform with the class of the input sample and at the same time ensure dissimilarity with input sample. Since the proposed scheme is differentiable it can be trained jointly ans in an end-to-end fashion with a classifier.

# Pros
- The text is well written in terms of the language and structure, while the authors adequately describe the proposed scheme.
- The contributions have been clearly stated
- The results along with the examples in Appendix C look promising

# Cons
- One of the claimed contributions (i.e. well suited for low-data regime) is not fully supported by the experimental results (Figure 2, CIFAR10 results)
- Some details are missing in the description of the experimental setup. Are the results presented in 3.1, with or without joint training with the classifier?
- In Figre 3 (section 3.2), there are no results with the classic baseline-light/strong DA. It would be insightful on how the joint training could affect the performance of the classifier.
- [minor] there some grammatical mistakes in the second paragraph of the Introduction

Overall, the text is easy to follow and well written. The combination of the spatial transformer block with a generative model is an interesting approach yet the performance seems to be slightly lower the state of the art. Considering also the fact that such generative schemes are difficult to define and train I wonder if it worth the effort. I would greatly appreciate such a discussion in the text.

---

### Official Review · AnonReviewer1 · 2019-04-07
**This paper proposed an interesting data augmentation method using GAN and STN, however the result is not so good.**

**Rating:** 3
**Confidence:** 1

**Review:**

The paper proposed to augment training data by spacial transformer network, which impose a set of simple transformations to the image and is fully differentiable. The method can learn data-specific transformations and outperforms other GAN-based methods in limited-data setting. Compared to Ratner, the method can be trained in an end-to-end manner, but the evaluation result is lower.

As the experiment result is not so good, I'd recommend a weak accept.

---

### Decision · Program_Chairs · 2019-04-15
**Acceptance Decision**

Accept